# NAP2: Neural Networks Hyperparameter Optimization Using Weights and Gradients Analysis

## Abstract

Recent hyper-parameter tuning methods for deep neural networks (DNNs) generally rely on first using low-fidelity methods to identify promising configurations and then using high-fidelity methods for further evaluation. While effective, existing solutions treat DNNs as 'black boxes', which limits their predictive abilities. In this work, we propose Neural Architectures Performance Prediction (NAP2), a 'white box' hyperparameter optimization approach. NAP2 models the changes in the weights and gradients of the analyzed networks over time and can predict their final performance with high accuracy, even after a short training period. Our evaluation shows that NAP2 outperforms the current state-of-the-art both in its ability to identify top-performing architectures and in the amount of resources it utilizes. Moreover, we show that our approach is transferable, meaning it is possible to train NAP2 on one dataset and apply it to another.

## 1 Introduction

Hyperparameter optimization (HPO) refers to a large array of techniques that aim to identify hyperparameter configurations that will optimize the performance of the analyzed algorithms Bischl et al. (2023). HPO approaches are diverse and include grid and random search Bergstra & Bengio (2012), evolutionary algorithms Li et al. (2013), Bayesian optimization Lindauer et al. (2022), and successive halving-based approaches Jamieson & Talwalkar (2016). Generally, all of the above approaches treat HPO as a black box optimization problem.

Performing HPO for deep neural architectures can be particularly challenging due to computational cost. Because training a large number of networks until convergence is often not practical, existing solutions Li et al. (2017); Jamieson & Talwalkar (2016) focus on the evaluation of intermediate results (i.e., partial training) and the dynamic allocation of resources to promising candidates. These approaches, which balance exploration and exploitation (e.g., multi-arm bandits Bouneffouf et al. (2020)), are the current state-of-the-art.

In this study, we propose Neural Architectures Performance Prediction (NAP2), a novel HPO approach for deep neural networks. Unlike existing approaches, NAP2 treats the neural architectures as white boxes and analyzes the changes in the weights and gradients of the networks during training. We model the weights and gradients using meta-features, which enable us to train a highly accurate prediction model that can identify promising candidate architectures in their early stages of training. Moreover, NAP2 can easily be used as part of existing HPO solutions (e.g., Successive Halving Jamieson & Talwalkar (2016)), as we show in our evaluation.

We evaluate NAP2 on two commonly-used datasets, CIFAR-10 and CIFAR-100, using architectures sampled from the NAS-Bench 101 dataset Ying et al. (2019). Our evaluation shows NAP2 outperforms SOTA baselines in both datasets, both in terms of accuracy and allocated resources. These results are even more significant given that we use the prediction model trained on CIFAR-10 on the CIFAR-100 architectures *without any fine-tuning*. These results show that NAP2 not only achieves top results but is also generic and transferable across datasets and architectures.

Our contributions in this study are as follows:

- We present NAP2, a novel HPO approach for neural networks. Our approach analyzes the inner workings of candidate architectures to predict their final performance and can achieve SOTA results. Additionally, our approach is transferable across datasets.

- We release our training dataset, which contains summary statistics of the weights and gradients of our training set networks at multiple points in their training process. This dataset is the first of its kind and can facilitate future research. We also make our code and trained prediction model publicly available.

## 2  RELATED WORK

### 2.1  HYPERPARAMETER OPTIMIZATION METHODS

All HPO methods reviewed in this section follow the same general approach: they iteratively generate hyperparameter configurations (HPCs) and evaluate them using a chosen metric. All configurations and results are saved in a repository, where they can be recalled as needed. In Bischl et al. (2023), the authors comprehensively review the field.

**Bayesian optimization (BO)**. BO Frazier (2018) is one of the most common approaches for HPO. It generally comprises of two components: a *probabilistic surrogate model* and an *acquisition function*. In each iteration, the surrogate model is fitted to all HPCs in the repository. Then, the acquisition model uses the distribution predicted by the surrogate model to select the next HPCs to be evaluated. Multiple variations of both the surrogate model and the acquisition function exist. The authors of McIntire et al. (2016) use sparse Gaussian processes as the surrogate model, while Eriksson et al. (2019) use local Bayesian optimization. SMAC3 Lindauer et al. (2022), one of the most commonly-used open-source optimization platforms, uses Random Forests. In recent years, neural networks have also been increasingly used as surrogate models Lim et al. (2021).

**Multifidelity-based approaches.** Multifidelity refers to approaches that use approximators with varying levels of fidelity. Low-fidelity approximators are less reliable in their predictions, but their computational cost is lower. Multifidelity-based HPO approaches initially use low-cost, low-fidelity approximators to evaluate many HPCs. The HPCs with the highest ranking are then further assessed using high-fidelity approximators. Two of the most well-known Multifidelity approaches are Successive Halving (SH) Jamieson & Talwalkar (2016) and Hyperband (HB) Li et al. (2017). SH is an iterative algorithm initialized with a total budget $B$ and predefined number of candidate HPCs $n$. In each iteration, the algorithm allocates a percentage of $B$ to the candidate HPCs, with the budget being divided equally. At the end of each iteration, the lowest-performing HPCs are discarded, thus enabling SH to further evaluate more promising candidates. HB attempts to allocate resources more efficiently by running multiple SH processes (referred to as brackets), each with a different number of HPCs. While all brackets have (roughly) the same budget, the varying number of HPCs enables the algorithm to experiment with different exploration-exploitation trade-offs.

### 2.2  NEURAL NETWORKS PERFORMANCE PREDICTION METHODS

**Lower fidelity methods.** One type of lower-fidelity methods uses deep reinforcement learning Tan et al. (2019); Baker et al. (2016) or evolutionary algorithm Real et al. (2019) to manage their resource allocation. An important difference from HPO methods is the *inability of these approaches to evaluate a fixed set of candidates*: NAS approaches explore a predefined search space, and the process is often stochastic. Moreover, these approaches often constrain the types of architectures they generate/evaluate (e.g., only residual blocks) to make the search problem more tractable.

**Learning curve extrapolation**. Methods that apply this approach attempt to train the neural network for a short time and then extrapolate the overall learning curve and final performance. Extrapolation is often performed using an ensemble of functions such as *pow3, log, and log-normal*. The work of Rorabaugh et al. (2021) hypothesizes that learning curves can be modeled with the function of the form: $f(x) = a - b^{(c-x)}$. This function is bounded with different constraints to only allow values possible by a learning curve. In Domhan et al. (2015), the authors use Markov Chain Monte Carlo inference to account for the uncertainty in the data and the network's parameters. This work is extended in Klein et al. (2017), where the authors use Bayesian neural networks.

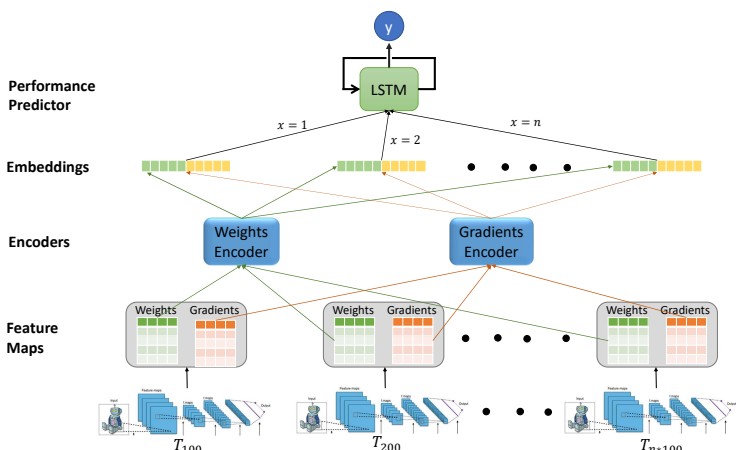

Figure 1: The proposed NAP2 architecture. At fixed batch intervals, we extract feature maps that represent the weights and gradients of the analyzed network. There feature maps are compressed using an encoder, and then provided as input to an LSTM model. This model predicts the analyzed network's final performance.

**Network topology-based prediction.** This line of work attempts to predict the performance of neural networks based on their structure. The Peephole architecture Deng et al. (2017) proposes to create an encoding for each analyzed network and then use logistic regression to predict its final performance on a given dataset. In Zhang et al. (2019), the authors use graph convolutional networks (GCNs) for the prediction task. A similar approach is used in Lukasik et al. (2021), where a graph neural network (GNN) is trained on the topology of the architectures of the NAS-Bench 101 Ying et al. (2019) dataset. While all previously reviewed works were trained for specific datasets, TAPAS Istrate et al. (2019) creates representations for both datasets and architectures to train its model.

## 3 THE PROPOSED METHOD

Our proposed approach is presented in Fig. 1. NAP2 is based on the hypothesis that analyzing the 'evolution' of neural architectures (i.e., changes in weights and gradients) in their early training stages will enable a learning model to predict their final performance. Starting with a set of $|A|$ architectures, we analyze every architecture by extracting summary statistics for each layer (Section 3.1). We repeat the process at fixed intervals of the network's training, thus obtaining multiple snapshots of its internal state. We represent the summary statistics using feature maps (Section 3.2) and sequentially provide them as input to an LSTM model that serves as our performance predictor (Section 3.3). Next, similarly to the successive halving used in Li et al. (2017), we discard a fixed percentage of $|A|$ whose performance is predicted to be the lowest (Section 3.4). We then train the remaining architectures for another set of steps and repeat the above process.

### 3.1 META-FEATURES EXTRACTION AND REPRESENTATION

#### 3.1.1 META-FEATURES EXTRACTION

We extract two sets of features for each analyzed neural architecture $A$: *gradients-based* and *weights-based*. The meta-features are extracted from each trainable layer—either dense or convolutional—at fixed intervals. In our experiments, we use 100 batches as our interval, a value we found to provide a good balance between the need for frequent snapshots of the analyzed architecture and the size of the log data that needs to be stored. All features are extracted in the same manner for each layer. Firstly, we calculate a single value on the entire layer's data (weights/gradients). Secondly, we calculate a more fine-grained view for each layer.

- For dense layers, we calculate the value for each neuron's inputs using the neuron's weights/gradients. For example, a layer of 100 neurons will produce meta-features consisting of 101 values each (1 for the entire layer and 100 for each neuron).

- For convolutional layers, we calculate the meta-features across kernels. For example, a convolutional layer with three input channels and 128 filters with a 4x4 kernel will produce $3 \times 4 \times 4 = 48$ values. Ultimately producing 49 features (1 for the entire layer and the 48 fine-grained features).

Once we obtain the sets of values for each of our analyzed layers, we can calculate our meta-features. These features are calculated in the same manner for the weights and gradient-based meta-features. We calculate the meta-features for each layer individually.

- **General statistics meta-features.** This group of meta-features consists of basic statistical operations: max, min, mean, variance, standard deviation, and median. Additionally, we calculate the quartile values for the 0, 25, 75, 50, 100 quartiles.

- **Distribution-based meta-features.** We calculate the co-variance, kurtosis and skewness of each analyzed layer.

$$\text{Skewness - } \frac{\mu_3}{\sigma^3} \tag{1}$$

$$\text{Kurtosis - } \frac{\mu_4}{\sigma^4} \tag{2}$$

  where $\mu_i = \frac{1}{N} \sum_k (w_{ij_k} - \overline{w_{ij}})^i$, $w_{ij}$ is the weight, $i$ is the index of the layer and $j$ is the index of the neuron, $\sigma$ is the standard deviation and $N$ is the number of weights of the current neuron.

- **Norm-based meta-features**. We calculate the $L_1$ and $L_2$ norms over the weights/gradients of each layer. In the following way: for a layer with weights $W$ the norms are:
  $L_1 : \ ||W||_1 = \sum_i abs(w_i) \ ; L_2 : \ ||W||_2 = \sqrt{\sum_i w_i^2}$.

Overall, we extract 24 meta-features per layer (12 for each of our two types). One challenge posed by our meta-features extraction method is their large variance in dimensioanlity across layers. We address this challenge in the following sections.

### 3.1.2 META-FEATURES REPRESENTATION

The main challenge in using meta-features to represent the inner processes of deep neural networks is the large variance of the latter in terms of topology. While neural architectures may differ significantly in width and depth, we require a fixed-size meta-features representation to train our performance prediction model. Therefore, we use a *feature maps-based representation* for our features. Feature maps (FM) enable us to represent layers and whole architectures using a fixed-size representation. Our process is carried out as follows:

- Because we create two sets of 12 meta-features per layer, we create two $[100, 12]$ matrices.

- For each meta-feature, we first insert the overall value (the summary value calculated over all neurons/filters) and then insert 99 randomly sampled neuron/filter-based values. In cases where there are fewer values, we use padding. For layers with a larger number of values, some information is discarded.

- If our matrices contain NaN values, we replace them with zeros. Inf+/- values are replaced with $+/-10^7$.

- We now stack individual layer representations to represent the overall architecture. For each neural net, we create two $[65, 100, 12]$ feature maps. This representation encapsulates the *first 65 layers* of the architecture. For smaller networks, we use padding; for larger architectures, we discard the final layers.

Upon completing this step, we will have created a fixed-size meta-features representation for each of our analyzed architectures, regardless of their topology. While this enables us to train a supervised learning model, the high dimensionality of our data still makes training difficult (consider we extract the feature maps every time we take a 'snapshot' of the network). In the following section, we address this problem.

**Input:** $M$, $F_i$, $C_i$, $C_d$, $S_i$, $F_s$
**Output:** Subset of selected top performing models
**begin**
    **Initialization** $W = \emptyset$, $G = \emptyset$

    **while** $|M| > F_S$ **do**
        $C_s \leftarrow$ Calculate new cut size

        $T_s \leftarrow$ Calculate new amount of train steps for agent iteration

        $W_i$, $G_i \leftarrow$ Train $M$ Models for $T_s$ steps and collect weight and gradients each $100^{th}$ step

        $S_w$, $S_g \leftarrow$ Calculate statistics on $W_i$ and $W_g$

        $M_w$, $M_g \leftarrow$ Generate feature maps on $S_w$ and $S_g$

        $W \leftarrow \text{insert}(M_w)$, $G \leftarrow \text{insert}(M_g)$

        $P \leftarrow$ Predict performance on $|M|$ models

        $BM \leftarrow |M| * C_s$ with worst performance

        $M \leftarrow M \setminus BM$

        iteration
    **end**
    **return** $M$
**end**

**Algorithm 1:** Selection Agent Algorithm. $W$ and $G$ denote the collected weights and gradients features maps for all current models, respectively. $C_s$ denotes the current iteration models' cut size. $T_s$ is the number of steps required to train all models in the current interation.

## 3.2 GENERATING FEATURE MAPS-BASED EMBEDDINGS

While informative, our feature maps have high dimensionality, which makes training the prediction model computationally expensive. A more condensed FM representation will enable us to deploy a smaller performance prediction model and use less training data. The latter advantage is also important because of the large computational cost of creating our FM-based dataset.

We use convolutional autoencoders (CAE) to create embeddings for each feature map. The architecture of the CAE is symmetric, and its dimensions are presented in Table 1 in the Appendix. We use the ReLU activation function and apply batch normalization after each layer. We use the MSE loss function and measure the distance between the original and reconstructed feature map using MSE as our loss function. The embedding layer is a vector $v$, where $|v| = 128$.

## 3.3 TRAINING THE PREDICTION MODEL

Our training process consists of two stages. First, we train two autoencoders, one for each type of feature map. We generate our training samples by randomly selecting architectures and time steps and retrieving the corresponding FM. In the second stage, we train our performance prediction model, an LSTM architecture with one hidden layer and an output layer with a Sigmoid function.

For each of our training set architectures, we train the model on sequences of FM in lengths of [1-10]. This setup means we train our model by providing it between one and ten sequential FMs (produced in 100-batch intervals) and then task the former with predicting the architecture's final performance. This form of training pushes our prediction model to produce accurate predictions based on limited information, which is more resource-efficient. We also use the maximal length sequence (i.e., 30 snapshots per sequence) to train our model to use all available information. Overall, we produce 11 training sequences per architecture.

As a result of our training process, we created a performance prediction model that is both generic and robust. The model is generic because we train it on multiple neural architectures with diverse topological features. This setup naturally forces our network to generalize. The robustness stems from the need to produce accurate predictions while facing large diversity in the amounts of provided information—from a single snapshot (i.e., only 100 batches) to 30.

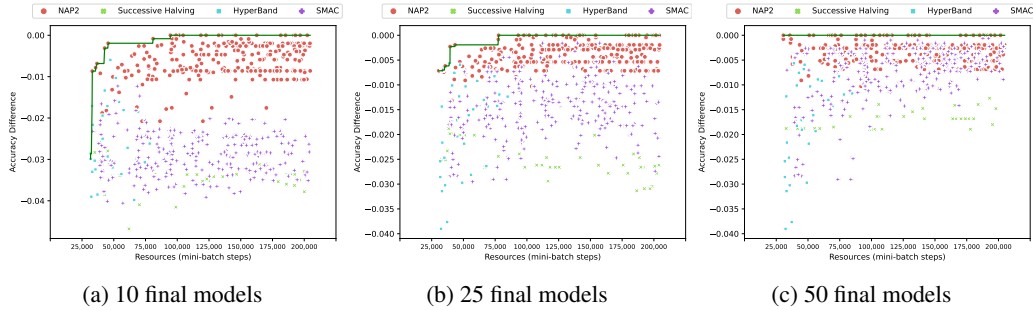

(a) 10 final models      (b) 25 final models      (c) 50 final models

Figure 2: Agents Accuracy Difference comparison on CNNS trained on Cifar-10, for different final number of selected models. The green line in each plot is the pareto frontier of the Accuracy Difference. Zero is the best result

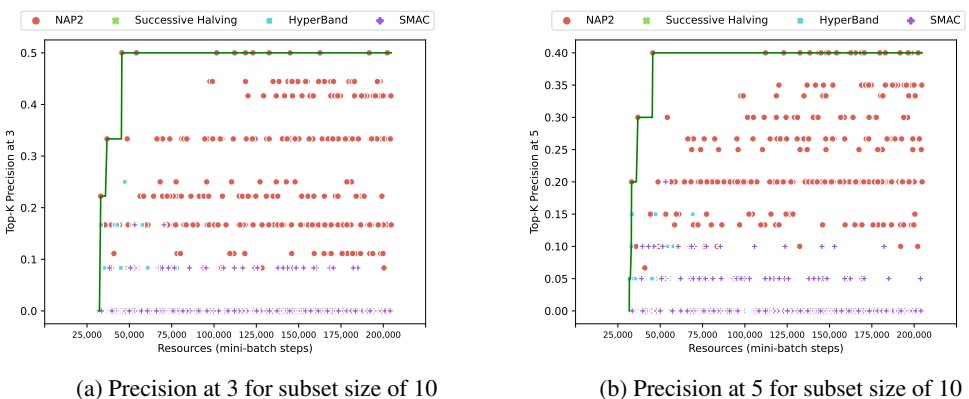

(a) Precision at 3 for subset size of 10      (b) Precision at 5 for subset size of 10

Figure 3: Precision-at-K comparisons using Cifar-10 CNNs. The green line in each plot is the pareto frontier of the Precision-at-K

### 3.4 ITERATIVE SELECTION OF MODELS

We use a straightforward method for model selection by building on Successive Halving Jamieson & Talwalkar (2016). At each step, we train all available architectures for a fixed number of steps (i.e., until the subsequent extraction of our meta-features). Next, we use our prediction model to predict the performance of all architectures and discard a fixed percentage of those with the lowest prediction.

Our approach is formally presented in Algorithm 1. NAP2 requires 6 inputs: **(1)** - $M$ – group of models, **(2)** $F_i$ – First iteration number of steps, **(3)** $C_i$ – starting cut size, **(4)** $C_d$ - Cut decay or increase rate, **(5)** $S_i$ – step interval in each agent iteration, **(6)** $F_s$ – Final number of models selected. Except for M, all other inputs affect the agent resource usage, thus affecting its accuracy.

## 4 EVALUATION

### 4.1 EXPERIMENTAL SETUP

**Training and meta-features extraction.** All architectures used in our experiments were randomly sampled from the NAS-Bench 101 dataset Ying et al. (2019), which consists of Inception-based architectures and their reported results. Because we need the analyzed networks' intermediary performance (for NAP2 and the baselines), we initialized and trained our sampled architectures. Due to our random initialization, performance may vary slightly from the one reported in Ying et al. (2019).

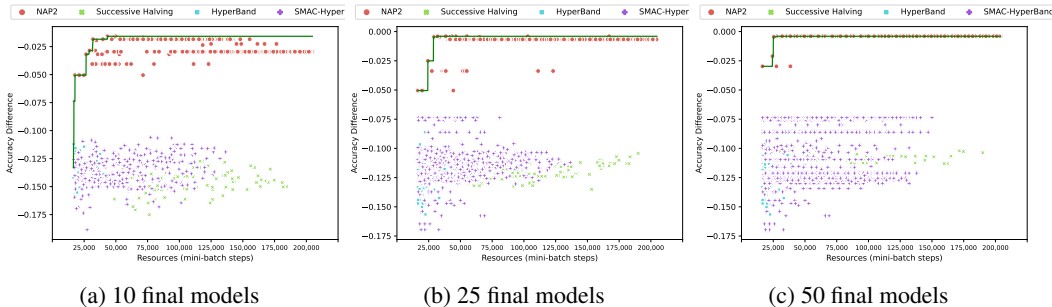

(a) 10 final models          (b) 25 final models          (c) 50 final models

Figure 4: Agents Accuracy Difference comparison on CNNS trained on CIFAR-100, for different final number of selected models. The green line in each plot is the pareto frontier of the Accuracy Difference. Zero is the best result

We use the results of our training throughout the remainder of this study. Our meta-features dataset, which is the first of its kind to be made public, is publicly available[1].

We trained 1,160 and 440 architectures on CIFAR-10 and CIFAR-100, respectively. All architectures were randomly sampled and trained for 36 epochs with a batch size of 128 (epoch=391 batches). The two architecture sets are disjoint. We replicate NAS-Bench 101's train/test/validation sets for training. For the architectures we ran on CIFAR-100, we modified the output layer to fit the dataset's dimensions. We extracted our feature maps every 100 batches (see Section 3.1), generating approximately four 'snapshots' per epoch. We generated 30 snapshots per architecture. Our training process resulted in two sets of architectures that are diverse both in performance and depth (see Table 2 and Fig. 7 in the Appendix).

We trained the autoencoders for 300 epochs (at a maximum) with early stopping. We use MSE as our loss function. Next, we trained the Performance Prediction Network on the train set folds while utilizing the validation set fold for early stopping. The training had a maximal duration of 150 epochs, with the $L_1$ loss function used on the differences in predicted network accuracy.

**Baselines' hyper-parameter configurations.** To ensure that our evaluation is extensive, *we ran every possible configuration of the following hyper-parameters* (values in brackets): a) Returned set Size (10, 25, 50); b) Mini-batches before cutoff (100, 500, 700, 1000); c) Successive halving cutoff rate (0.1, 0.15, 0.2, 0.25, 0.35, 0.45, 0.5, $dynamic$); d) The $\eta$ parameter (all integers in the range $[2, 29]$); e) Initial design of N configurations (1, 5, 10, 20, 30, 50, 75, 100, 150, and 200); f) Number of trials (25, 50, 75, 100, 125, 150, 200, and 250). A full deception of the hyper-parameters is presented in the Appendix.

## 4.2 EVALUATION METRICS

We use four-fold cross-validation for both datasets, with reported results averaged across all folds. We use three metrics to evaluate the performance of our proposed approach:

**Resources used.** We defined one unit of resources to correspond to 100 mini-batch steps of a single CNN during training (equivalent to $\sim \frac{1}{4}$ epoch). Note that we can disregard the hardware used in our experiments using this definition.

**Top-K precision.** For a final set of architectures returned by the algorithm, we calculate the percentage of top-K-performing architectures included in the set. For example, when calculating the top-3 precision for a final set of 10 architectures, the presence of the two highest-performing architectures yields a precision of 66%.

**Accuracy difference from the top performer.** We measure the accuracy differences between the chosen architectures and the top-performing architecture on the relevant dataset. This measure complements top-k precision because it enables the detection of cases where the returned architectures are not in the 'top', but their performance is worse by a tiny margin.

---

[1]Due to its size, we cannot share the dataset anonymously. It will be made available pending acceptance

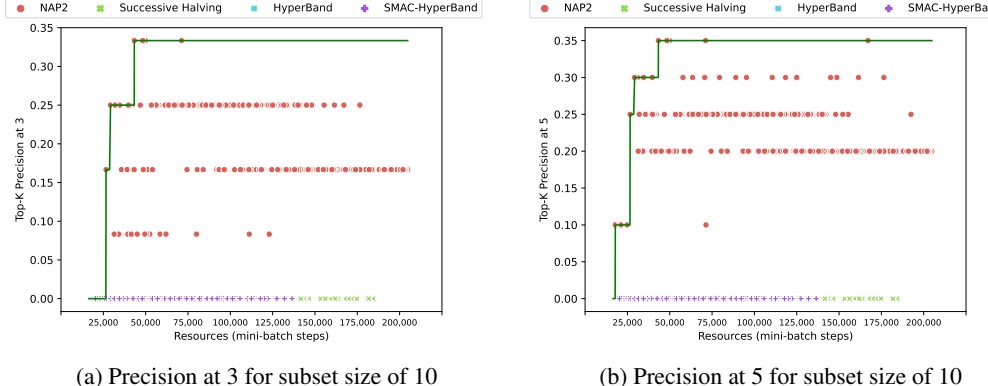

(a) Precision at 3 for subset size of 10    (b) Precision at 5 for subset size of 10

Figure 5: Precision-at-K comparisons using CIFAR-100 CNNs. The green line in each plot is the Pareto frontier of the Precision-at-K

## 4.3 BASELINES

We compare NAP2 to the following baselines:

**Successive Halving** Jamieson & Talwalkar (2016). An iterative approach that trains the neural architectures in each iteration for a pre-specified amount of mini-batch steps. SH evaluates the networks' validation losses at each iteration and discards the worst performers according to a predetermined cut-off parameter.

**Hyperband** Li et al. (2017). HB builds upon the Successive Halving algorithm. It conducts multiple SH runs with different parameter configurations, designed to evaluate a varying number of architectures using different training times. The multiple runs are carried out using different neural architectures, selected uniformly from our predefined set of architectures.

**SMAC3.** SMAC3 Lindauer et al. (2022) is a highly-cited and popular hyperparameter optimization framework. SMAC3 uses Bayesian optimization with the Hyperband algorithm to improve its performance. We use HPObench Eggensperger et al. (2021) to define our search space and map each of the algorithm's sample selections to the most relevant architecture.

Both Hyperband and SMAC3 are configured to return only a single network. For our evaluation, we require a ranked list of architectures, so we modified these two algorithms to return multiple networks. The networks were selected based on their most recent performance.

## 4.4 EVALUATION RESULTS

### 4.4.1 CIFAR-10 EVALUATION RESULTS

**Accuracy difference from top performer.** We plot the performance of all evaluated configuration, NAP2 and the baselines, in Fig. 2. We use three different final set sizes (i.e., the number of architectures chosen as the algorithm's output): 10, 25, and 50. In all experiments, NAP2 significantly outperforms all the baselines, and forms the Pareto Front. Another key conclusion is that NAP2 requires a very limited budget to reach top performance: we only needed to run each evaluated architecture for a few batch-iterations to include a top-performing architecture in our final set.

**Top-K precision as a function of resource usage.** Fig. 3 presents the results for the *Top-3* and *Top-5* architectures for a final sets of 10 architectures. The results for 25 architectures are presented in the Appendix. NAP2 again outperforms the baselines in all experimental setups. Note that in the more challenging setup—Top-3 precision—NAP2 is always able, on average, to detect at least one top-performing architecture, unlike the baselines.

The results clearly demonstrate the advantages of our white-box approach, compared to existing black-box approaches. An additional analysis of HyperBand (the top-performing baseline), is presented in the Appendix.

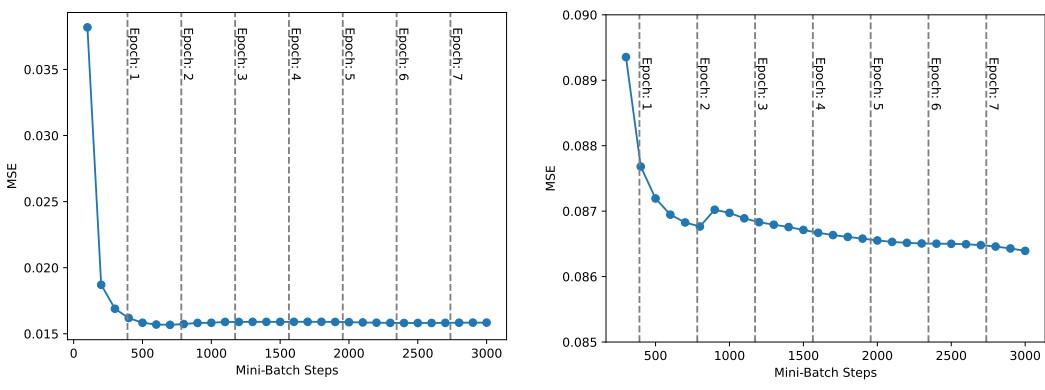

(a) MSE error of the model on CIFAR-10 test sets

(b) MSE error of the model on CIFAR-100 dataset

Figure 6: The MSE error of NAP2 on CIFAR-10 and CIFAR-100 as a function of the overall number of mini-batch steps. For CIFAR-100, we use the model trained on CIFAR-10 (no additional training).

### 4.4.2    CIFAR-100 EVALUATION RESULTS

In the previous section, NAP2's train and test architectures were trained on the same dataset. We now evaluate our approach's transferability by using the prediction model trained on CIFAR-10 to predict the performance of previously unseen architectures on the CIFAR-100 dataset.

We repeat the experiments presented in Section 4.4.1. We run our experiments on four 110-architecture folds and report the average results. Fig. 4 presents our results for the accuracy difference metric. It is clear that NAP2 is again the top performer, with a much lower accuracy difference than all evaluated baselines. Moreover, for the larger final model set sizes (25 and 50), NAP2's accuracy difference is nearing zero, and the gap between our approach and the baselines is much larger than in the CIFAR-10 experiments.

Next, we analyze the results for the Top-K precision metric. Figure 5 presents the results for top-3 and top-5 accuracy for two final set sizes: 10 and 25. The results once again show that NAP2 significantly outperforms the two baselines. It is evident in all experiments that even the worst-performing NAP2 architecture outperforms all baseline configurations.

### 4.5    ANALYSIS: PERFORMANCE PREDICTION ERROR OVER TIME

Until now, we evaluated NAP2's ability to identify top-performing architectures. We have not, however, evaluated the accuracy of its predictions (i.e., how close our predictions were to the actual performance). We now calculate the mean squared error (MSE) of all our predictions—both for CIFAR-10 and CIFAR-100—and plot them as a function of training duration.

Our results are presented in Fig. 6. NAP2 is highly accurate, with an MSE error smaller than 0.1 for both datasets. It is also clear that our model requires very little training data - the performance of our method plateaus after one training epoch for CIFAR-10, and two epochs for CIFAR-100. These results show that our approach can produce highly accurate predictions at a minimal training budget.

## 5    CONCLUSION.

We present NAP2, a novel performance prediction model for neural networks. Unlike existing solutions, our approach analyzes the 'inner workings' of networks and can, therefore, predict their performance with high accuracy. Our evaluation shows that NAP2 requires analyzing only a fraction of a network's initial training to produce a highly accurate prediction. Our evaluation demonstrates the superiority of our approach over the current state-of-the-art.

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

## A ADDITIONAL DETAILS ON THE PROPOSED METHOD AND THE EVALUATION

### A.1 THE DIMENSIONS OF OUR CONVOLUTIONAL AUTOENCODER

Table 1: CAE Layers' Dimensions

| Layer num. | Layer Dimensions |
| --- | --- |
| Layer 1 | (1, 100, 512) |
| Layer 2 | (65, 1, 256) |
| Layer 3 (embedding layer) | (1, 1, 128) |
| Layer 4 | (65, 1, 256) |
| Layer 5 | (1, 100, 512) |

### A.2 NEURAL ARCHITECTURES AND HYPER-PARAMETERS CONFIGURATIONS DETAILS

We diversify our two architecture datasets, to test the ability of NAP2 and the baselines to generalize over multiple hyperparameter configurations. We randomly assigned the follwoing options to our architectures: Adam and SGD optimizers (NAS-Bench 101 only uses the latter), and multiple options for the learning rate. When using the Adam optimizer, we disabled the cosine decay used in Nas-Bench 101. We also allowed for networks to run without layer regularization, as in the original Nas-Bench code.

We experiment with multiple configurations of the following hyperparameters:

- **Returned set Size.** The number of architectures returned as a final output. We experimented with set sizes of 10, 25, and 50.

- **Mini-batches before cutoff.** The number of mini-batch steps each of our evaluated architectures is trained prior to another iteration of the baselines and NAP2. We used sizes of 100, 300, and 500.

- **Initial CNN Steps.** The number of steps all evaluated architectures are trained prior to the first cutoff. We experimented with different hyperparameters for NAP2 and the evaluated baselines. We used size of 100, 500, 700, and 1000. Note that for SMAC3, we controlled this parameter by changing the scenario minimum budget.

- **Successive Halving cutoff rate.** The cutoff rate determines the percentage of architectures that are discard with each operation of the baseline. We experimented with the following values: 0.1, 0.15, 0.2, 0.25, 0.35, 0.45, and 0.5. Additionally, we enabled the baseline to automatically set this value (the decision is made dynamically based on the number of architectures and available resources).

- **The $\eta$ parameter.** We experimented with multiple values of $\eta$ for the Hyperband baseline. This hyperparameter determines the percentage of discarded architectures in each of the algorithm's multiple successive halving iterations. We refer the reader to Li et al. (2017) for full details. In our experiments, we experimented with all integers in the range $[2, 29]$. This range consists of all the legal values for $\eta$ in our experimental setup. For SMAC, we set this value to 3, as suggested in its documentation.

- **Initial Design N Configurations.** This hyperparameter is only relevant for SMAC3, as it set the number of configuration used in the first iteration of the algorithm. we experimented with multiple values: 1, 5, 10, 20, 30, 50, 75, 100, 150, and 200.

- **Number of Trials.** The number trials used by SMAC algorithm. To allow for comparison with other methods we ignored the wall clock time restrictions, and used only the number of trials to limit the SMAC algorithm steps. Values used: 25, 50, 75, 100, 125, 150, 200, and 250.

## A.3 THE DEPTH AND ACCURACY DISTRIBUTION OF THE EVALUATED ARCHITECTURES

The results presented in Figure 7 show the performance of all architectures when trained to convergence. Table 2 presents the number of layers distribution of the evaluated architectures.

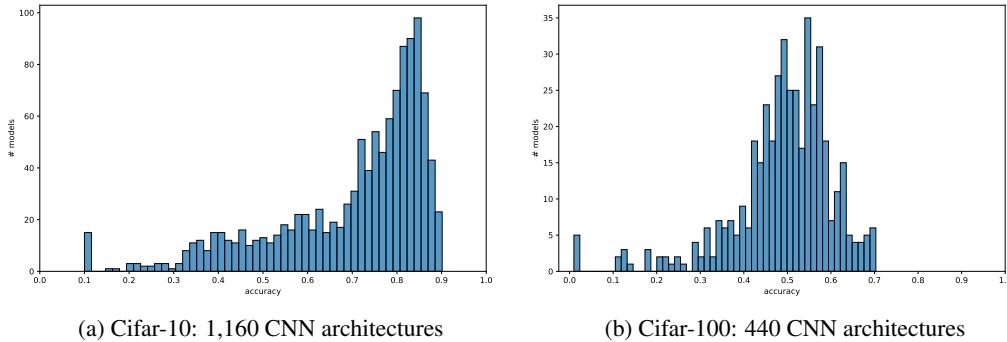

(a) Cifar-10: 1,160 CNN architectures  (b) Cifar-100: 440 CNN architectures

Figure 7: Histograms presenting the accuracy distribution of our evaluated CNN architectures in two datasets: CIFAR-10 and CIFAR-100.

|  | Depth≤30 | Depth≤40 | Depth≤50 | Depth≤60 |
|---|---|---|---|---|
| CIFAR-10 | 27 | 212 | 473 | 448 |
| CIFAR-100 | 3 | 58 | 223 | 215 |

Table 2: CNN Architectures Depth by Dataset

## A.4 THE TOP-K PERFORMANCE METRIC RESULTS FOR A FINAL SET OF 25 ARCHITECTURES

These results, presented in Figures 8 and 9, show the performance of NAP2 and the baselines for a larger final set. As in the previous results, NAP2 significantly outperforms the baseline.

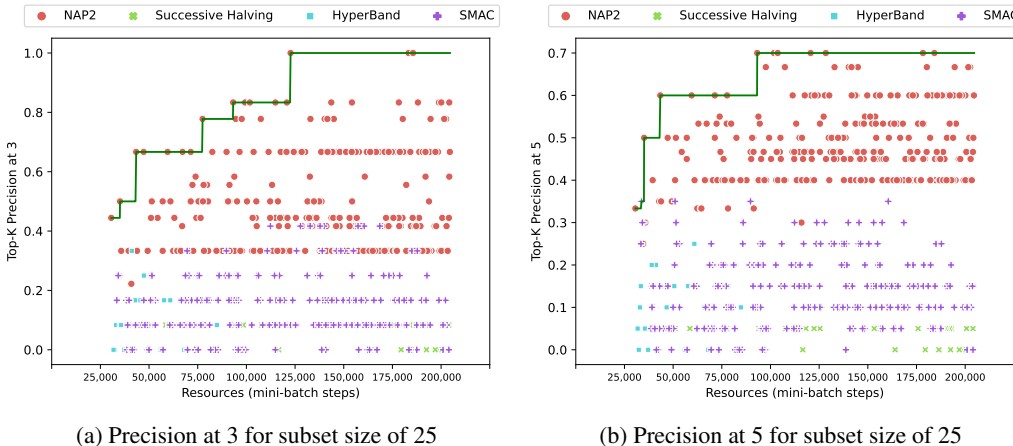

(a) Precision at 3 for subset size of 25      (b) Precision at 5 for subset size of 25

Figure 8: Precision-at-K comparisons using Cifar-10 CNNs. The green line in each plot is the Pareto frontier of the Precision-at-K

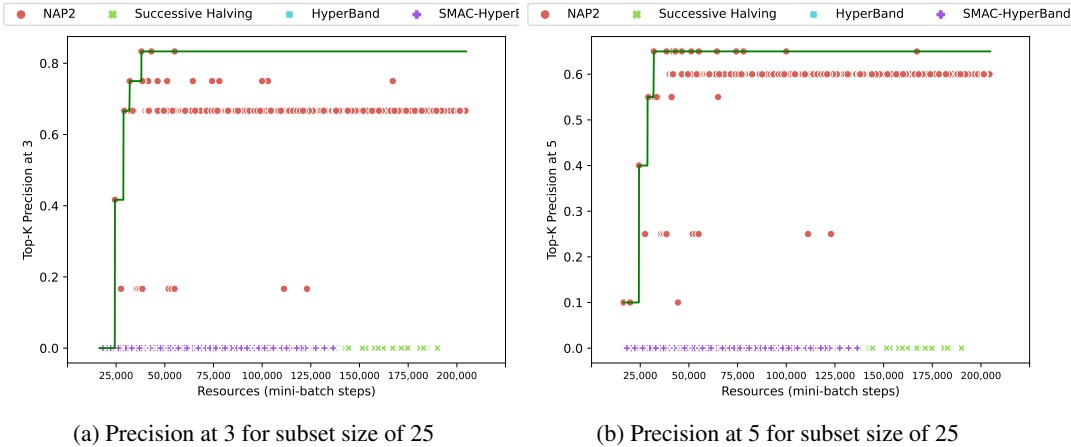

(a) Precision at 3 for subset size of 25      (b) Precision at 5 for subset size of 25

Figure 9: Precision-at-K comparisons using CIFAR-100 CNNs. The green line in each plot is the Pareto frontier of the Precision-at-K

## A.5 ADDITIONAL ANLYSIS OF THE BASELINES' PERFORMANCE

**Analyzing the differences between HB configurations.** While it is clear that NAP2 outperforms HB by a wide margin overall, Fig. 2 shows that some HB runs were able to achieve results that are close to those of our approach, for some budgets. An analysis of the top-performing HB configurations showed that many of them had $\eta = 3$, which is one of the recommended configurations in the original paper. However, it is important to note that *two thirds* of the $\eta = 3$ configurations fared poorly in the accuracy difference metric. Moreover, when analyzing HB performance with respect to the top-k metric, less than 50% of $\eta = 3$ configurations were able to identify a top-performing architecture for any $k$. It is clear that HB can achieve comparable results to NAP2, but only when the optimal configurations are known in advance (which is, of course, not possible).

