# OpenReview forum: "NAP2: Neural Networks Hyperparameter Optimization Using Weights and Gradients Analysis"
_ICLR.cc/2024/Conference — Submitted to ICLR 2024_

### Official Review · Reviewer_W1Sa · 2023-10-15

**Soundness:** 3 good
**Presentation:** 2 fair
**Contribution:** 2 fair
**Rating:** 5
**Confidence:** 4

**Summary:**

This paper presents an approach to do hyperparameter optimization by building a predictive model based on the evolution of weights as features within the neural network. For each architecture decisions on which architecture is useful are based on the evolution of weights or features extracted from weights during the training process. Decision making based on these features is used using typical MAB methods such as HyperBand. Validation on this proposed approach is provided on CIFAR-10 and CIFAR-100 dataset.

**Strengths:**

The paper combines many different techniques in order to deliver empirical results.

The paper uses a somewhat new approach of inspecting the weights in order to make decisions on whether the architecture is useful or not.

The paper performs validation on CIFAR-10 and CIFAR-100 showing that their approach works in this setting.

**Weaknesses:**

The paper is quite incremental.

The paper does not cover many related works in this area which have used similar techniques. See [1], [2], [3] among others.

Even if the above related work did not exist, it is not clear to the reviewer whether the contribution of the paper is sufficiently novel or useful to give a score of accept.

The paper's validation on CIFAR-10 and CIFAR-100 may not be sufficient evidence for it to be shown to be useful in practice on larger neural networks than toy networks.

[1] Freeze-thaw Bayesian optimization. Kevin Swersky, Jasper Snoek, Ryan Prescott Adams. https://arxiv.org/abs/1406.3896.

[2] When to Prune? A Policy towards Early Structural Pruning. Maying Shen, Pavlo Molchanov, Hongxu Yin, Jose M. Alvarez. Proc. CVPR 2022.

[3] Unifying and Boosting Gradient-Based Training-Free Neural Architecture Search. Yao Shu, Zhongxiang Dai, Zhaoxuan Wu, Bryan Kian Hsiang Low. Proc. NeurIPS 2022.

**Questions:**

What do the authors believe is a reasonable path to move this paper towards some form of acceptance given the above concerns?

---

> ### Author Response · Authors · 2023-11-13
> **Response to Reviewer W1Sa**
>
> We would like to begin by thanking the reviewer for their time and effort. Below we address the weaknesses raised by the reviewer.
>
> 1) **"The paper does not cover many related works in this area which have used similar techniques. See [1], [2], [3] among others. Even if the above related work did not exist, it is not clear to the reviewer whether the contribution of the paper is sufficiently novel or useful to give a score of accept."**
>
> The reviewer provided three representative studies from different domains (HPO, pruning, and NAS). We argue that NAP2 has significant novelties compared to these studies, and those similar to them. We will briefly describe each study, then elaborate on our method’s novelties.
>
> [1] is a relatively early (from 2014) HPO study, that analyzes the convergence of the training curves throughout training to predict final performance. The method uses Bayesian Optimization to decide which architectures are worthy of additional resources. In essence, this approach is a precursor of current SOTA methods that use similar principles (the methods that are our baselines). It is important to note that the architectures are not treated as `white-boxes’, and that there is no modeling of the weights/gradients of individual layers.
>
> [2] is a study in the field of network pruning. Unlike the majority of studies in the field, this study attempts to prune the network before it is fully trained. Similar to the large majority of pruning studies, the authors analyze the weights of individual neurons in the layer to determine which ones need to be pruned (the search is guided by sub-network similarity). Decisions are reached based on neuron weight and sub-network similarity.
>
> [3] is NAS study that analyzes the gradients of architectures to efficiently predict their performance. The gradients are evaluated at the architecture level, thus treating the neural net as a black box.
>
> Compared to these studies (and those similar to them), NAP2 contains multiple novelties:
>
> *a)* We are the first to analyze changes in weights and gradients **over time**, and show that it is highly effective for the task of HPO.  We are the first to analyze the evolution of the architecture throughout its training. Furthermore, this is the first time that such a white-box approach has been proposed to the tasks of HPO (or NAS, to the best of our knowledge).
>
> *b)* Unlike [2] and other pruning-based approaches that analyze weights using a simple heuristic, we develop a set of features that capture various traits of each layer. Additionally, we are the first to analyze the gradients at the layer level rather than the architecture as a whole. Additionally, we propose ways to represent this information using feature maps and compress it so that it is more usable.
>
> *c)*	Unlike all other learning-based solutions (in the domains of HPO, NAS, and pruning), we are the first to propose a model that can be trained on one set of architectures, then applied on another disjoint set. As shown in our experiments, our model is transferable also to architectures that are trained on other datasets. Because our model is reusable, our approach is highly efficient in an online setting (one only needs to extract the feature maps once every 100 mini-batches).
>
> 2) **"The paper's validation on CIFAR-10 and CIFAR-100 may not be sufficient evidence for it to be shown to be useful in practice on larger neural networks than toy networks."**
>
> We conduct our evaluation on architectures sampled from the NAS-Bench 101 dataset, which is the same dataset used in all leading SOTA studies. Given that, we follow the standard evaluation in this domain.
>
> The evaluated architectures are by no means “toy networks”: as shown in Table 2, we evaluate NAP2 on hundreds of architectures with more than 30 layers, with hundreds more above 50 layers.
>
> **Conclusion:** The reviewer requested that we elaborate on our contributions and the robustness of our evaluation. We analyzed the studies mentioned by the reviewer, and explained that our work has novel aspects that they do not have: a) temporal analysis of weights and gradients; b) multiple features that represent various aspects; c) transferability between architectures and datasets. Additionally, we explained that our evaluation is based on the dataset used for evaluation in current state-of-the-art.
>
> We will be happy to answer any other questions the reviewer may have.

---

> > ### Comment · Reviewer_W1Sa · 2023-11-13
> > **Revising my score**
> >
> > Hmm,
> >
> > The authors make a persuasive argument from the perspective of empirical implementation. Much of my research work is at the intersection of Category Theory and Computational Complexity, which may make me not the ideal reviewer for this work.
> >
> > I will revise my score, but I am not sure whether I can argue for acceptance.

---

> > > ### Author Response · Authors · 2023-11-13
> > > **RE: Revising my score**
> > >
> > > We thank the reviewer for reading and responding to our rebuttal, and for revising the score upwards.
> > >
> > > If the reviewer has any other reservations regarding our paper  (novelty, experimentation, etc.), we will be happy to address them to the best of our abilities.

---

### Official Review · Reviewer_p8Rh · 2023-10-31

**Soundness:** 3 good
**Presentation:** 2 fair
**Contribution:** 2 fair
**Rating:** 3
**Confidence:** 4

**Summary:**

Different from the existing black-box HPO method that treats DNNs as black boxes, this paper proposes a method called NAP2 that treats DNNs as white boxes. NAP2 predicts final performance from weights and gradients of neural networks. The authors claim that the proposed method improves the performance over the baselines.

**Strengths:**

It is interesting that the final performance of a neural network training can be predicted by the information of its weights and gradients the learned predictors are transferable to other datasets.

**Weaknesses:**

The experiments are limited to neural architecture search only on CIFAR-10 and CIFAR-100, while the authors claimed that it is a hyperparameter optimization method. As they do not provide any theoretical perspective of this approach, they need more experiments in terms of tasks and datasets to support their claim.
    Gradient-based hyperparameter optimization, e.g., [1], can also be regarded as a white-box approach. Are there any comparisons with this line of works?

[1]: Maclaurin et al. "Gradient-based Hyperparameter Optimization through Reversible Learning," ICML 2015.
Questions:

**Questions:**

The paper only presents mini-batch steps, but how long does it take for architecture search in wall-clock time?

---

> ### Author Response · Authors · 2023-11-13
> **Response to Reviewer p8Rh**
>
> We would like to begin by thanking the reviewer for their time and effort.
>
> 1) **"The experiments are limited to neural architecture search only on CIFAR-10 and CIFAR-100"**
>
> Our experiments are conducted on the NAS-Bench dataset, which is the current standard for HPO and NAS-based tasks (see recent studies from both domains in [1], [2]). NAS-Bench 101 contains large and complex architectures with dozens of layers (that are trained on CIFAR). We refer the reviewer to Table 2 in our paper for information about the architectures’ sizes.
>
> Given that this dataset is used by practically every study, we maintain that our choice of dataset is justified.
>
> 2) **"As they do not provide any theoretical perspective of this approach, they need more experiments in terms of tasks and datasets to support their claim."**
>
> We use the NAP2 prediction model in a Successive Halving setup [3]. As such, we can rely on the same theorem-backed performance guarantees as this algorithm (see Section 3.1 in the relevant paper). Additionally, we compare NAP2 to well-known and widely-used state-of-the-art baselines, and outperform them by highly significant margins.
>
> 3) **"Gradient-based hyperparameter optimization, e.g., [1], can also be regarded as a white-box approach. Are there any comparisons with this line of works?”**
>
> The study mentioned by the reviewer is NOT a white-box approach. That study uses gradient descent to modify the hyperparameters used for training of an architecture (e.g., the Momentum optimization value), then trains the architecture until convergence. This means that this approach, *like all other methods we review and compare NAP2 to*, treat the architecture as a black-box. They do not analyze weights, gradients, or their evolution over time. As we explain in our study, no existing HPO or NAS approaches analyze the weights and gradients in a similar way to NAP2.
>
> Moreover, a comparison to this line of work is not consistent with the current state-of-the-art. While policy gradient-based approaches were used in the past for HPO and NAS (see the highly influential work by [4]), they have a prohibitively high computational cost when applied in scale. This led to the adoption of other, more efficient techniques, the most common of which being Bayesian optimization.
>
> 4) **"The paper only presents mini-batch steps, but how long does it take for architecture search in wall-clock time?"**
>
> Our per-architecture running time is nearly identical to those reported by our baselines, because all methods are evaluated on NAS-Bench 101 dataset architectures. The only additional computational cost of NAP2 compared to the baselines at test time is the extraction of the meta-features once every 100 mini-batches (which is efficient because we sample values from each layer), and using the already-trained NAP2 model to predict the architecture’s performance.
>
> When considering the computational cost, it is important to differentiate between the computational cost of NAP2 during training and evaluation. The training of the NAP2 requires computational resources because we train multiple architectures to convergence, extract meta-features and the train the prediction model. This process, however, can be done `offline’, and on a different set of architectures and datasets than those used for the evaluation (which is our experimental setup). At test time, the meta-features only need to be extracted once every 100 mini-batches, and the small pre-trained prediction model is used once to produce a prediction (requires less than a second). In a sense, our trained model can be thought of as a LLM: studies in the field of natural language processing use extremely large models trained by others (e.g., OpenAI), and do not consider the training of those models because they were done by other.
>
> [1] Mellor, Joe, et al. "Neural architecture search without training." International Conference on Machine Learning. PMLR, 2021.
>
> [2] Wistuba, Martin, Arlind Kadra, and Josif Grabocka. "Supervising the multi-fidelity race of hyperparameter configurations." Advances in Neural Information Processing Systems 35 (2022): 13470-13484.
>
> [3] amieson, Kevin, and Ameet Talwalkar. "Non-stochastic best arm identification and hyperparameter optimization." Artificial intelligence and statistics. PMLR, 2016.
>
> [4] Zoph, Barret, and Quoc V. Le. "Neural architecture search with reinforcement learning.",2016.

---

> > ### Comment · Reviewer_p8Rh · 2023-11-22
> > **Response to the authors**
> >
> > Thank you for the response, and I am sorry for my delayed reply.
> >
> > On 1 and 2, I still think that benchmarking only on two datasets limits the contribution of this work.
> >
> > On 3, to my understanding, black-box optimization is derivative-free optimization, and, contrarily, white-box optimization is optimization with gradient information, such as gradient descent. If so, gradient-based HPO methods fell into white-box HPO. Gradient-based HPO is used in the NAS literature, for example, DARTS by Liu et al.
> >
> > On 4, to my knowledge, presenting search cost in GPU hours and/or wall clock time is common in the NAS literature.

---

### Official Review · Reviewer_Q7He · 2023-10-31

**Soundness:** 3 good
**Presentation:** 2 fair
**Contribution:** 2 fair
**Rating:** 5
**Confidence:** 5

**Summary:**

This paper proposes Neural Architecture Performance Predictor (NAP2), which predicts final accuracy of neural network by analyzing dynamics of weights and gradients of network. NAP2 uses meta-features of neural network’s weights and gradients. NAP2 is utilized in neural architecture search by combining with Successive Halving. With sample architectures from NAS-Bench-101, NAP2-based NAS outperform than other hyperparameter optimization algorithms.

**Strengths:**

To the best of my knowledge, NAP2 is the first approach that trying to use meta-features of both weights and gradients. Also, the author’s hypothesis that “analyzing the ‘evolution’ of neural architectures (i.e., changes in weights and gradients) in their early training stages will enable a learning model to predict their final performance.” is reasonable.

**Weaknesses:**

The proposed method is more like a 'performance predictor' than a 'hyperparameter optimization method'[1]. NAP2 is trained with more than 1,000 evaluation results of fully trained neural network samples, but the comparison methods did not use them. It is more reasonable to compare it with performance predictors like BANANAS[2], NPENAS[3], or AG-Net[4].
(Since I do not have access to the dataset, I assume that the accuracy of the model is between 0 and 1 based on the paper's report that they use a sigmoid function for the output of the predictor.) Figure 6 shows that the mean square error between performance prediction and final accuracy is about 0.015, which is larger than the mean absolute error of other performance prediction results[5] in NAS-Bench-101 (mean absolute error < 0.01). Also, the result MSE > 0.86 in CIFAR-100 doesn't substantiate NAP2’s transferability if the scale of accuracy is 0 to 1.

[1] White, C., Zela, A., Ru, R., Liu, Y., & Hutter, F. (2021). How powerful are performance predictors in neural architecture search?. Advances in Neural Information Processing Systems, 34, 28454-28469.

[2] White, C., Neiswanger, W., & Savani, Y. (2021, May). Bananas: Bayesian optimization with neural architectures for neural architecture search. In Proceedings of the AAAI Conference on Artificial Intelligence (Vol. 35, No. 12, pp. 10293-10301).

[3] Wei, C., Niu, C., Tang, Y., Wang, Y., Hu, H., & Liang, J. (2022). Npenas: Neural predictor guided evolution for neural architecture search. IEEE Transactions on Neural Networks and Learning Systems.

[4] Lukasik, J., Jung, S., & Keuper, M. (2022, October). Learning where to look–generative nas is surprisingly efficient. In European Conference on Computer Vision (pp. 257-273). Cham: Springer Nature Switzerland.

[5] Siems, J. N., Zimmer, L., Zela, A., Lukasik, J., Keuper, M., & Hutter, F. (2020). Nas-bench-301 and the case for surrogate benchmarks for neural architecture search.

**Questions:**

I wonder if statistics of weights and gradients can represent the neural network without their connectivity. If those have meaningful information, similarly embedded networks will show similar performance. Is there experiment or evidence that supporting this hypothesis, like arch2vec[6] shows? (e.g. correlation between predicted performance and final accuracy, visualization of embedding vectors, etc.)
NAS-Bench-201[7] reports that the best architecture on CIFAR-10, CIFAR-100, ImageNet16-120 are slightly different. I also think the optimal architectures of different dataset should relate with complexity of dataset. Why NAP2 is “generic and transferable across datasets and architectures”?

[6] Yan, S., Zheng, Y., Ao, W., Zeng, X., & Zhang, M. (2020). Does unsupervised architecture representation learning help neural architecture search?. Advances in neural information processing systems, 33, 12486-12498.

[7] Dong, X., & Yang, Y. (2020). Nas-bench-201: Extending the scope of reproducible neural architecture search. arXiv preprint arXiv:2001.00326.

---

> ### Author Response · Authors · 2023-11-12
> **Response to Reviewer Q7He**
>
> We would like to begin by thanking the reviewer for their time and effort.
>
> We first address the weaknesses raised by the reviewer. This is particularly important because the **reviewer has MISREAD** the scale  on Figure 6 (0.86 instead of 0.086), which led to a serious misunderstanding on the reviewer’s part as to the performance of our approach (see #2):
>
> 1) **“It is more reasonable to compare it with performance predictors like BANANAS[2], NPENAS[3], or AG-Net[4]”**
>
> While the reviewer is correct in stating that previous studies (Bananas, Npeans, etc.) used performance prediction, we believe the reviewer overlooked a *critical point*: all mentioned studies require the learning model to be trained on the evaluated architectures. *NAP2, on the other hand, can be trained on one set of architectures, then applied to another set* – even one trained on another dataset (CIFAR-100 in our case) without any additional training.
>
> This difference has significant implications: first, we can train the prediction model `offline’, without any connection to the test set architectures. Then, we can use the trained model in multiple experiments, even on other datasets. Secondly, for our model to work at test time, we only need to extract our meta-features once every 100 mini-batches (the interval size we used in our evaluation). Coupled with the fact that we sample values from each layer, this process has negligible computational cost. This means that our setup is the same as the HPO baselines to which we compare: we begin our evaluation with a fixed set of architectures on which we have no knowledge.
>
> 2) **"Figure 6 shows that the mean square error between performance prediction and final accuracy is about 0.015, which is larger than the mean absolute error of other performance prediction results[5] in NAS-Bench-101 (mean absolute error < 0.01). Also, the result MSE > 0.86 in CIFAR-100 doesn't substantiate NAP2’s transferability if the scale of accuracy is 0 to 1.”**
>
> We would like to begin with the second point: the **reviewer has MISREAD Figure 6**: the MSE error for CIFAR-100 is 0.086, NOT 0.86. This order of magnitude makes all the difference in understanding our results, since they show that a model that was trained on CIFAR-10 architectures is transferable to CIFAR-100.
>
> Secondly, the reviewer is correct that [5] achieved an error of <0.01, but that was achieved after multiple epochs of training and the use of several training seeds. Our approach, on the other hand, achieves MSE of about 0.015 AFTER A SINGLE EPOCH on the evaluation set architectures.  Additionally, as shown in Figures 2-5, NAP2 requires less than an epoch (100-300 mini-batch steps) to significantly outperform all the baselines.
>
> **Summary:** To conclude our response to the weaknesses raised by the reviewer, our model is different from NAS-oriented solutions in critical ways: out model is trained offline (without any connection to the evaluated architectures/search space), it is transferable to other architectures and datasets, and its white-box setup enables it to produce highly accurate predictions after than less of epoch of training, compared with tens of epochs required by existing solutions (for example, the curve-based methods mentioned in [1] require at least a full epoch of training to get any data points).

---

> > ### Author Response · Authors · 2023-11-12
> > **Response to Reviewer Q7He's questions**
> >
> > **Response to the Reviewer’s questions:**
> > 1) **"I wonder if statistics of weights and gradients can represent the neural network without their connectivity..."**
> >
> > Our approach also considers the networks topology, but  we use a simpler representation of the architectures than the one described in [6]. As explained in Section 3.1.2, we create two sets of 65 feature maps, and use them to represent the first 65 layers of each analyzed architecture. The ordering of the feature maps is based on the architecture’s topology, and we use padding when the network’s depth is below 65.
> >
> > Our representation differs from previous work because our focus is on the evolution of the architecture’s parameters over time, rather than its topology. Moreover, because we sample values from each layer, similarity in features does not necessarily indicate similar topology. We used t-SNE on our embedding representations, but were not able to detect any discernible patterns. It is possible that more advanced explainability methods would be more effective (particularly those designed for temporal data), but this is beyond the scope of our current work.
> >
> > 2) **"NAS-Bench-201[7] reports that the...I also think the optimal architectures of different datasets..."**
> >
> > There is a key difference between our work and [7]: the latter analyzed tens of thousands of architectures, and thus were able to reach broad conclusions. Our experiments were conducted on 1,500 architectures that were sampled from NAS-Bench-101. Because of the smaller sample size,  we observe different performance distribution:  in our experiments, some of our top-performing architectures had 30+ layers, while others had around 60.
> >
> > Because of the relatively small sample sizes, we also cannot form a conclusion regarding the complexities of architectures and their respective datasets.
> >
> > 3) **"Why NAP2 is “generic and transferable across datasets and architectures”?"**
> >
> > As we explain in our response to the weaknesses detailed by the reviewer, the NAP2 model is trained on one set of architectures, then applied without any additional learning to another. We conduct two sets of experiments:
> >
> > a)	Training on one set of CIFAR-10-trained architectures, then applying the model on another (disjoint) set of architectures trained on the same dataset.
> >
> > b)	Training the model on CIFAR-10-trained architectures, then applying the model on a disjoint set of architectures trained on a different dataset (CIFAR-100).
> >
> > As shown in our results (see Figures 2-5), NAP2 significantly outperforms the baselines in both experiments. Moreover, it is clear that our approach only requires 100-300 mini-batches (less than one epoch) to identify top-performing architectures.
> >
> > It is likely that the reviewer’s question stems from misreading the MSE values in Figure 6 (0.86 instead of the actual value of 0.086). As mentioned above, NAP2 was able to obtain these values after approximately one epoch of training, compared to dozens required by previous studies dealing with prediction.

---

> > ### Comment · Reviewer_Q7He · 2023-11-15
> >
> > Thanks for the reply of the authors.
> >
> > 1. Thanks to the author's detailed answer, I fully understand the importance of offline learning nature of NAP2 and how it utilizes the evolution of parameters over time.
> > 2. I sincerely apologize for the confusion. The 0.86 mentioned in the review was a typo, and I was correctly recognizing that the value in Figure 6(b) is 0.086. However, based on the accuracy distribution in Figure 7(b), 0.086 is likely higher than the MSE of "simply predicting the accuracy of all models to be 0.5" (most candidate models have CIFAR-100 accuracy between 0.3 and 0.7, so most individual squared errors are smaller than 0.04). Therefore, I am concerned that NAP2's predictions are likely to rely on the CIFAR-10 accuracy scale, and my concerns about NAP2's generic performance are not yet resolved.
> > To substantiate claims about the generic performance of NAP2, the authors should demonstrate NAP2's ability to overcome these scale differences or establish that these differences are inconsequential to NAP2's performance. For instance, the authors could present a scatterplot or report a rank correlation illustrating the relationship between NAP2 predictions and CIFAR-100 accuracy for individual models, thereby providing a more robust evaluation of NAP2's predictive capabilities for CIFAR-100.
> > I will keep my original rating for now.

---

> > > ### Author Response · Authors · 2023-11-15
> > > **Re: Comment by Reviewer Q7He**
> > >
> > > We thank the reviewer for the useful suggestion! The use of ranking correlation is certainly more informative than only reporting the MSE.
> > >
> > > To test the ranking correlation between the ground truth and the rankings produces by NAP2, we used the Kendall Tau correlation test. Using this test, we compare our ranked list of architectures to the “ground truth” (the rankings of the architectures based on their actual final performance).
> > >
> > > We evaluated the rankings produced by our model after 100 and 200 training mini-batches (i.e., right before discarding the first and second batches of architectures).  We ran the test on each of the four-folds we use for evaluation in CIFAR-10, and for all the architectures of CIFAR-100.
> > >
> > > For CIFAR-10, our average coefficient for 100 mini-batches was 0.365 with a standard deviation of 0.055. For 200 mini-batches, the correlation was 0.322 with a standard deviation of 0.061. For CIFAR-100, our correlation coefficients were 0.298 and 0.303 for 100 and 200 mini-batches, respectively. All correlations were statistically significant with p<0.01. While the results for CIFAR-100 are lower than those for CIFAR (as expected), all results are clearly correlative.
> > >
> > > These additional results complement the results of Figures 2-5, which present our method’s performance for the final set of selected architectures. By implementing the reviewer’s useful suggestion, we are able to conclude that NAP2 is not only able to consistently identify a small set of high-performing architectures but is actually able to generalize well over the entire population of architectures. Additionally, as shown by our results on CIFAR-100, NAP2 is able to predict the relative performance of architectures (i.e., ranking), even when the performance distribution of the population varies.
> > >
> > > We will revise our manuscript to include the results of the new experiments.
> > >
> > > If the reviewer has any additional questions or reservation regarding our work, please let us know. We will do our best to address them.

---

> > > > ### Comment · Reviewer_Q7He · 2023-11-17
> > > >
> > > > Thanks to the authors for their efforts.
> > > >
> > > > The rank correlation values reported by the authors show that NAP2 trained on CIFAR-10 can also produce a reasonable metric for accuracy on CIFAR-100.
> > > > However, the reported correlation seems to be worse than the SOTA training-free performance estimator [1].
> > > >
> > > > Since they are using completely different methodologies (especially as NAP2 is able to track changes in parameters over time), it may not be fair to make a direct comparison. Nevertheless,  theoretical background and the additional experiments provided are not significant enough to support the author's hypothesis.
> > > >
> > > > The concerns I had about NAP2's generic performance have been partially addressed, but I will maintain my original score.

---

> > > > > ### Author Response · Authors · 2023-11-17
> > > > >
> > > > > The reviewer is once again comparing apples and oranges. As we've mentioned before, the performance estimators mentioned by the reviewer require training the analyzed networks for multiple epochs. Our approach achieves the reported performance after 100-200 mini-batches.
> > > > >
> > > > > More generally: hyper-parameter optimization, as its name suggests, is about finding the optimal configuration (or one that is as close as possible). We have consistently shown that our approach significantly outperforms current SOTA in finding such architectures.

---

### Official Review · Reviewer_2Mh1 · 2023-11-01

**Soundness:** 2 fair
**Presentation:** 2 fair
**Contribution:** 2 fair
**Rating:** 3
**Confidence:** 5

**Summary:**

This paper introduces NAP2, a hyperparameter optimization method for deep neural networks. Unlike existing approaches that treat DNNs as 'black boxes,' NAP2 is a 'white box' method that accurately predicts DNN performance by modeling changes in weights and gradients. NAP2 outperforms current state-of-the-art methods and is transferable across datasets, making it a valuable advancement in hyperparameter tuning for DNNs.

**Strengths:**

The paper introduces a method to predict model performance based on meta information and its feature map and gradients overtime. The idea is well-motivated. The method could potentially help improve hyperparameter optimization by reducing evaluation cost if the prediction of performance can be done with a small number of training budget.

The authors evaluate their methods on CIFAR-10 using the NAS-Bench 101, and show how it can be transfered to CIFAR-100.

**Weaknesses:**

While the idea is great, there are a few critical issues with the whole experiment desgin.
1. If the focus of the method is for hyperparameter optimization, then the evaluation should follow a HPO pipeline, i.e., there should be no training dataset of models with ground-truth performance. The whole experiment should be in an online fashion. Otherwise, the proposed method is just a prediction model and do not show its usability in HPO.
2. The evaluation of resources used is unclear to me. Is the cost of training and infering NAP2 counted in calculation? Other baseline methods such as Successive Halving do not require extra computation, so you should probably compensate for those methods with extra computation to make it a fair comparison.
3. On CIFAR-100, the comparison methods seem to perform extremely bad, e.g., none of them get any precision > 0. I wonder if the configurations are reasonable according to the original work? It is unlikely that these methods should use the same set of configurations.
4. For figure 6, is it saying that the prediction model trained on CIFAR-10 is able to predict on CIFAR-100 with MSE error < 0.1? This is a bit surprising because the scale of accuracy is quite different, i.e., on CIFAR-10 the models can achieve 0.9 accuracy but should be much lower like 0.6 on CIFAR-100. I wonder how is the proposed method handle this difference in scale?
5. In Figure 2-5, it seems that all 3 methods do not benefit with more resources, which is not intuitive. I wonder if the authors could include a simple hill climbing or BO baseline to better illustrate the results?

Other weaknesses:
1. The figures are hard to read. Please increase the font sizes.

**Questions:**

See weaknesses.

---

> ### Author Response · Authors · 2023-11-12
> **Response to Reviewer 2Mh1**
>
> We would like to begin by thanking the reviewer for their time and effort. Below we address the weaknesses raised by the reviewer.
>
> 1) **Weakness #1**
>
> The reviewer touched on a point that is critical to understanding the strength and re-usability of our model. The reviewer would be correct if the prediction model had to be trained anew for each task. This is **NOT** the case: our prediction model can be trained once, then used in other prediction tasks. We demonstrate the re-usability of our trained model in two ways, with varying degrees of difficulty:
>
> a) Training our model on a set of architectures that were trained CIFAR-10, and then performing HPO on a another (disjoint) set of architectures trained on the same dataset.
>
> b) Training our model on a set of architectures that were trained CIFAR-10, then performing HPO on a set of architectures that were trained on CIFAR-100.
>
> While the second setup is more difficult for our model, because of the need to generalize, NAP2 performs very well in both scenarios.
>
> It is precisely for the reasons mentioned above, that we include in our contributions (see Section 1) the release of the trained model along with our dataset and code. The inclusion of the model is designed to enable other researchers to utilize our approach more easily.
>
> We will update our paper to better emphasize this point.
>
> 2) **Weakness #2**
>
> The cost of training our model is not included in our evaluation. Our reasoning is as follows:
>
> a) As explained in our response to the previous comment, because the idea behind our model is “train once, use multiple times” (including on other datasets), we do not consider the training cost to be relevant to the evaluation.
>
> b) The training of the NAP2 model can be done ‘offline’, a long time before the online evaluation.
>
> c) Conceptually, our work is similar to studies that use pre-trained LLMs for various tasks. These LLMs require huge amounts of time and compute resources, but these costs are not factored into the evaluation, because they are used `as-is’. The same logic applies to the training of our model.
>
> We will update our paper to better emphasize this point.
>
> 3) **Weakness #3**
>
> The explanation for the baselines’ poor performance on CIFAR-100 is simple, and comprises of three reasons:
>
> a)	The number of architectures is larger: in CIFAR-10 we have 1160 architectures but use a four-fold evaluation. In CIFAR-100, we have 440 architectures, and all are used for evaluation. The larger number of architectures makes detecting the top performers  more difficult (esp. a small number like top-3 or top-5).
>
> b)	Performing well on CIFAR-100 is more challenging because of the larger number of classes (see Figure 7 in the appendix for performance distribution). As a result, the baselines (which do not have NAP2 predictive capabilities) perform more poorly.
>
> c)	We show the results for small number of epochs, because NAP2 does not need more in order to perform well. We elaborate on this further in our response to question #5 below.
>
> It should also be noted that none of our baselines used the top-3/5 precision in their reported evaluation. We include it in our evaluation specifically because it is a very challenging task that only NAP2 manages to do well.
>
> 4) **Weakness #4**
>
> Because our goal was to create a generic and re-usable model, we did not make any adaptations to our prediction model: we trained it on CIFAR-10 architectures, then evaluated it on the CIFAR-100 architectures. Our model was able to achieve good results on CIFAR-100, which strongly supports our claim of our model being generic. However, Figure 6 shows that compared to CIFAR-10, our model is far less accurate on CIFAR-100: after one epoch, our model’s MSE is x4.5 higher, and after seven epochs the difference is x5. This lower accuracy is also reflected in our results (Figures 2-5), where NAP2 underperforms on CIFAR-100 compared to CIFAR-10 (but still far better than the baselines).
>
> As stated by the reviewer, the difference in architecture accuracy distribution likely contributed to the different accuracy. The different training dynamics because of the need to classify x10 labels is likely a larger contributing factor.
>
> 5) **Weakness #5**
>
>  The reason for the lack of improvement by the baselines is simple: 100 mini-batches are about a quarter of an epoch (see Section 4.2). For CIFAR-10, we use four-fold cross validation, so one epoch for all evaluated architectures (before pruning architectures) is ~110,000 mini-batches. This means that we present results for about four epochs of training.
>
> This small number of epochs is more than enough for our approach, which begins to be effective after as little as 100 mini-batches (a quarter of an epoch). This is the result of our approach being a white-box approach, and its ability to obtain insights mid-epoch (as opposed to, for example, curve extrapolation prediction methods). The other approaches are too slow to succeed in such a short time frame.

---

> > ### Comment · Reviewer_2Mh1 · 2023-11-23
> >
> > Thanks to the authors for the reply. After reading the rebuttal, I have a few thoughts that may help to further improve this work towards a publication.
> >
> > > The cost of training our model is not included in our evaluation. Our reasoning is as follows:
> >
> > > a) As explained in our response to the previous comment, because the idea behind our model is “train once, use multiple times” (including on other datasets), we do not consider the training cost to be relevant to the evaluation.
> >
> > > b) The training of the NAP2 model can be done ‘offline’, a long time before the online evaluation.
> >
> > > c) Conceptually, our work is similar to studies that use pre-trained LLMs for various tasks. These LLMs require huge amounts of time and compute resources, but these costs are not factored into the evaluation, because they are used `as-is’. The same logic applies to the training of our model.
> >
> > This is absolutely not true. If the method aims to perform HPO, it is possible that the effort used to train this NAP2 would have obtained better results by performing, e.g., grid search, directly on the target search task. Making a connection between this specific method with foundation models is unreasonable unless the authors could demonstrate the generalization of their method on multiple large-scale benchmarks, at least at the level of ImageNet.
> >
> > Also, for weakness 4, I would like to clarify that I misread the MSE and thought it was RMSE, which I realized after reading the comments from Reviewer Q7He. I sincerely apologize for this.
> >
> > However, with that being said, an MSE of 0.08 on accuracy prediction is not going to work anyway, since a reasonable guessing like taking a mean would have been better than that. For example, if you take two sets of random numbers from uniform U(0,0.7), the MSE between them would be around 0.08.
> >
> > Due to this issue, I think the results are not supportive of the authors's claim, and I unfortunately have to downgrade my score.

---

### Meta-Review · Area_Chair_CzF6 · 2023-12-03

**Metareview:**

The paper addresses the important problem of hyper-parameter optimization from a white-box perspective, utilizing weights and gradients to train performance predictors. The method, although incremental in terms of novelty, is methodologically principled.

Unfortunately, reviewers raised important points on the weaknesses of the proposed work, concerning the experimental protocol, the impact of the results, and its incremental nature. On the other hand, they appropriately criticized conducting NAS experiments with only CIFAR-10 and CIFAR-100.

I agree with the reviewers. In its current state, it is premature to derive conclusive statements on the predictive performance of the technique. As a result, I recommend rejection. The authors are encouraged to consider the reviewers' comments and improve the work's empirical aspect for the next submission.

**Justification For Why Not Higher Score:**

The experimental protocol is limited and the results are not conclusive.

**Justification For Why Not Lower Score:**

N/A

---

### Decision · Program_Chairs · 2024-01-16

Reject